# Rats with Long-Term Cholestasis Have a Decreased Cytosolic but Maintained Mitochondrial Hepatic CoA Pool

**DOI:** 10.3390/ijms24054365

**Published:** 2023-02-22

**Authors:** Lukas Krähenbühl, Stephan Krähenbühl

**Affiliations:** 1Visceral Surgery, Hospital Leuggern, 5316 Leuggern, Switzerland; 2Clinical Pharmacology & Toxicology, University Hospital Basel, 4031 Basel, Switzerland

**Keywords:** long-term cholestasis, free CoA (CoASH), acetyl-CoA, hippurate excretion, N-acetylsulfamethoxazole excretion, palmitate activation

## Abstract

Previous studies showed that rats with long-term bile duct ligation have reduced coenzyme A stores per g of liver but maintained mitochondrial CoA stores. Based on these observations, we determined the CoA pool in the liver homogenate, liver mitochondria, and liver cytosol of rats with bile duct ligation for 4 weeks (BDL rats, *n* = 9) and sham-operated control rats (CON rats, *n* = 5). In addition, we tested the cytosolic and mitochondrial CoA pools by assessing the metabolism of sulfamethoxazole and benzoate in vivo and of palmitate in vitro. The hepatic total CoA content was lower in BDL than CON rats (mean ± SEM; 128 ± 5 vs. 210 ± 9 nmol/g), affecting all subfractions equally (free CoA (CoASH), short- and long-chain acyl-CoA). In BDL rats, the hepatic mitochondrial CoA pool was maintained, and the cytosolic pool was reduced (23.0 ± 0.9 vs. 84.6 ± 3.7 nmol/g liver; CoA subfractions were affected equally). The urinary excretion of hippurate after i.p. benzoate administration (measuring mitochondrial benzoate activation) was reduced in BDL rats (23.0 ± 0.9 vs. 48.6 ± 3.7% of dose/24 h), whereas the urinary elimination of N-acetylsulfamethoxazole after i.p. sulfamethoxazole administration (measuring the cytosolic acetyl-CoA pool) was maintained (36.6 ± 3.0 vs. 35.1 ± 2.5% of dose/24 h BDL vs. CON rats). Palmitate activation was impaired in the liver homogenate of BDL rats but the cytosolic CoASH concentration was not limiting. In conclusion, BDL rats have reduced hepatocellular cytosolic CoA stores, but this reduction does not limit sulfamethoxazole N-acetylation or palmitate activation. The hepatocellular mitochondrial CoA pool is maintained in BDL rats. Impaired hippurate formation in BDL rats is explained best by mitochondrial dysfunction.

## 1. Introduction

The importance of coenzyme A (CoA) in energy metabolism has been established since the early studies by Lipman et al. [1]. In particular, CoA is essential for the activation of carboxylic acids to thioesters and is, therefore, as free CoA (CoASH) or in its acylated form, indispensable for many metabolic processes. For example, within mitochondria, fatty acid β-oxidation and Krebs cycle activity rely on CoASH, and in the cytosol, acetyl-CoA is needed for cholesterol and fatty acid synthesis. In mammals, CoA is synthesized by all cells with a nucleus in a series of five reactions from pantothenate (vitamin B5), cysteine, and ATP [2,3]. The first three steps, catalyzed by pantothenate kinases, phosphopantothenoylcysteine synthetase, and phosphopantothenoylcysteine decarboxylase, are located in the cytosol [4]. The last two steps are catalyzed by the bifunctional enzyme coenzyme A synthetase, whose subcellular localization is currently debated and may be on the outer mitochondrial membrane and/or in the mitochondrial matrix. The pantothenate needed is provided by food (mainly in the form of CoA) and by intestinal bacteria [4]. To keep the CoA pool constant, not only CoA synthesis is important, but also degradation. While CoA 3′-dephosphorylation can be performed by acid phosphatases in the lysosomes, the hydrolyzation of CoA to phosphopantetheine is performed by NUDT7 and NUDT19 (nudix (nucleoside diphosphate linked moiety X)-type motif) in peroxisomes and NUDT8 in mitochondria [4]. The resulting phosphopantetheine can be used to resynthesize CoA. CoASH and acyl-CoAs are polar molecules that cannot pass biological membranes by diffusion [5,6,7]. Transport into cellular compartments (for example, mitochondria and peroxisomes) is, therefore, performed by transporters such as SLC25A42 for mitochondria [8] and SLC25A17 for peroxisomes [9]. These transporters act as exchangers, exchanging CoA for, e.g., ATP, ADP, AMP, or FAD [4]. Within cells, CoA is compartmentalized, with important pools within mitochondria, peroxisomes, and the cytosol. In hepatocytes, the highest concentrations are located within mitochondria (in the low millimolar range), whereas the concentrations in the cytosol and in the peroxisomes are approximately 10 times lower [4].

In previous studies, we have shown that rats with long-term cholestasis due to ligation of the common bile duct (BDL rats) have a reduced hepatic total CoA content expressed per g liver [10] whereas the CoA content in isolated liver mitochondria was maintained [11]. The decrease in the hepatic CoA content concerned not only CoASH but also all acyl-CoA fractions determined [10], suggesting a decrease in CoA synthesis and/or an increase in CoA degradation. Considering the importance of the CoA pool for hepatic metabolism and function, we decided to investigate the mitochondrial and cytosolic CoA pools in BDL rats both quantitatively and functionally. For that, we determined the CoA pool in isolated liver mitochondria, in a cytosolic fraction, and in total liver homogenate of BDL rats and tested the metabolism of benzoate as a marker of the mitochondrial CoASH pool and sulfamethoxazole and palmitate as markers of the cytosolic acetyl-CoA and CoASH pools, respectively.

## 2. Results

The rats used in the current study are characterized in Table 1. While the body weight was not different between the two groups, BDL rats had higher liver and spleen weights, higher serum bilirubin and bile acid concentrations, and higher activities of AST and alkaline phosphatase.

The results of the fractionation of the liver homogenate are shown in Appendix A. In total (mitochondrial and cytosolic fraction), we isolated 61% and 54% of the protein contained in the homogenate in CON and BDL rats, respectively. The losses result from discarding the nuclear pellet and the light portion of the pellet after the 7600× *g* centrifugations, as described in Methods. Per g of liver, we isolated less mitochondrial protein and less citrate synthase activity in BDL compared to CON rats, which reflects the reduced hepatocyte volume fraction (due to ductular proliferation and fibrosis) in BDL rats [12]. The total mitochondrial content per g of liver (isolated mitochondria corrected by the recovery) was not different between BDL and CON rats. In both groups, the contamination of the mitochondrial preparation with cytosolic protein was negligible. Regarding the cytosolic fraction, the recovery of lactate dehydrogenase was approximately 90% in both groups and the total cytosolic protein (isolated protein corrected by recovery) was not different between BDL and CON rats. The mitochondrial contamination was larger in BDL compared to CON rats but was not more than 2% in both groups.

The hepatic CoA content is displayed in Table 2. The total CoA content was reduced by 39% in BDL compared to CON rats. This reduction concerned all CoA fractions determined, namely CoASH by 31%, SCA-CoA by 40% and LCA-CoA by 31%. Accordingly, the percentage of SCA-CoA in relation to total CoA was in the range of 65% and not different between the two groups.

The CoA content in liver mitochondria expressed per mg of mitochondrial protein is given in Figure 1. As can be seen, there were no differences between the two groups, neither in the total CoA content nor in the subfractions. When expressed per g of liver (Table 2), the total mitochondrial CoA content was reduced by 12.4% in BDL compared to CON rats, which reached statistical significance. A numerical reduction was observed in all subfractions, reaching statistical significance for CoASH and LCA-CoA. The contribution of the mitochondrial to total CoA was significantly higher in BDL compared to CON rats (77 ± 4% vs. 53 ± 4%).

The reduction in the total hepatic CoA content in BDL rats could mainly be explained by a reduced CoA content in the cytosol (Table 2). Compared to CON rats, the total cytosolic CoA content was reduced by 80% in BDL rats. A reduction was observed for all subfractions, amounting to 80% for CoASH, 70% for SCA-CoA, and 92% for LCA-CoA.

Since we had observed a small decrease in the mitochondrial CoA content when expressed per g of liver (but not per total liver, if the liver weight is considered) and a pronounced decrease in the cytosolic CoA content (in particular also for the SCA-CoA fraction) in BDL compared to CON rats, we decided to investigate the possibility that these decreases affect metabolic pathways relying on CoASH (mitochondria) or CoASH and acetyl-CoA (cytosol). As shown in Figure 2, we determined the urinary excretion of hippuric acid after the administration of benzoic acid for assessing the mitochondrial CoASH pool. Benzoic acid needs to be activated to the corresponding CoA derivative in the mitochondrial matrix before it can react with glycine to hippurate [13], which is excreted in the urine. As shown in the figure, after the i.p. administration of benzoate, hippurate excretion over 24 h was significantly reduced by 37% in BDL compared to CON rats (30 ± 3% vs. 48 ± 4% of the dose administered in BDL vs. CON rats).

To investigate the cytosolic acetyl-CoA pool, we quantified the urinary excretion of acetyl-sulfamethoxazole (acetyl-SMX) after the i.p. administration of SMX (Figure 3). As can be seen in the figure, the excretion of acetyl-SMX over 24 h was not different between the two groups (35 ± 3% vs. 37 ± 3% of the administered dose in BDL vs. CON rats).

To test whether the CoA concentration in the hepatocyte cytoplasm is limiting for palmitate metabolism, we investigated the activation of palmitate (formation of palmitoyl-CoA) and the formation of acid-soluble products from palmitate (mostly β-hydroxybutyrate [11]) by liver homogenate and isolated mitochondria at different CoA concentrations. Considering the assay for liver homogenate (5 mg tissue in a final volume of 500 µL) and the cytosolic CoASH concentration (Table 2), the estimated CoASH concentration in the assay without the addition of exogenous CoASH was 0.2 µM in control and 0.04 µM in BDL rats. We determined palmitate activation and the formation of acid-soluble products from palmitate without the addition of exogenous CoASH and at CoASH concentrations of 5 µM (corresponding to the hepatocellular cytosolic concentration in BDL rats) and 150 µM. As shown in Table 3, in liver homogenate without the addition of exogenous CoASH, the activities of palmitate activation and acid-soluble product formation were significantly lower than in the presence of 5 or 150 µM CoASH. However, the activities at 5 µM were not different from those at 150 µM CoASH for control and BDL rats, suggesting that the cytosolic CoASH concentration was not limiting the metabolism of palmitate in either group of rats. Further, in the presence of exogenous CoASH, the activities were 30–40% lower for palmitoyl-CoA activation and 60–70% lower for acid-soluble product formation for BDL compared to control rats. Regarding isolated mitochondria, the activation of palmitate was not different between BDL and control rats whereas the formation of acid-soluble products was approximately 30% lower in BDL compared to control rats.

To explain the observed decrease in hippurate formation and the impaired hepatic palmitate metabolism in BDL rats, we investigated the oxidative metabolism of different substrates by isolated liver mitochondria. As shown in Table 4, the state 3 oxidation of glutamate and succinate was impaired in mitochondria from BDL compared to control rats, indicating decreased activity of the mitochondrial electron transport chain. In comparison, RCR and ADP/O ratios were not significantly different between BDL and control rats.

## 3. Discussion

The study confirms our previous findings that the hepatic CoA content (expressed per g liver) is reduced in BDL compared to control rats, whereas the mitochondrial CoA content (expressed per mg of mitochondrial protein) is maintained [10,11]. We could show directly that the reduction in the hepatic CoA content in BDL rats is due to a drop in the cytosolic CoA pool. In contrast to the changes observed in the respective subcellular CoA pools, the N-acetylation of sulfamethoxazole (a cytosolic reaction needing acetyl-CoA) was maintained, whereas the formation of hippurate from benzoate (a mitochondrial reaction needing CoA) was reduced in BDL compared to control rats. The activation of palmitate was impaired in BDL rats in liver homogenate but not in isolated mitochondria, whereas the formation of acid-soluble products was impaired in BDL compared to control rats in both liver homogenate and mitochondria.

In sham-operated control rats, we determined a hepatic CoA content of 210 µg/g liver wet weight, which is in the lower range compared to other studies [2,4] but close to the findings of a study with a similar work-up of liver tissue [14]. When comparing the liver CoA content between studies, it is important to consider whether the animals studied were in fed or in fasted states since the hepatic CoA content increases with starvation [15,16]. In the current study and in the study by Brass and Ruff [14], the animals were investigated in the fed state, explaining why the hepatic CoA content was in the lower range of the reported values.

In the current study, the mitochondrial CoA content was in the range of 1.5 nmol/mg of mitochondrial protein (Figure 2), again in agreement with the study by Brass and Ruff [14]. Considering a mitochondrial volume fraction of 20% in hepatocytes [12,17], the mitochondrial total CoA concentration can be estimated to be in the range of 0.5–1 mM, which is lower than the approximately 5 mM reported in the review article by Naquet et al. [4]. Again, the feeding state of the animals may play an important role. Based on the results of the current study and assuming a cytosolic volume fraction of 0.60 in hepatocytes [17], the total CoA concentration in the cytoplasm would be in the range of 150 µM for control rats and 50 µM for BDL rats, which agrees with the concentrations reported by Naquet et al. [4]. Since SCA-CoA was the main fraction of the total cytosolic CoA (this study) and acetyl-CoA is the main hepatic SCA-CoA [10], these values approximately reflect the cytosolic acetyl-CoA concentration. The large gradient between the mitochondrial and the cytosolic CoA concentration underscores the tightness of the inner mitochondrial membrane towards CoASH and acyl-CoAs and suggests that the transport of CoASH and acyl-CoAs from the mitochondrial matrix into the cytoplasm is very limited.

Since CoASH and acyl-CoAs cannot diffuse across biological membranes and, therefore, cannot leave cells [5,6,7], the findings of the current study must be explained by changes in the hepatocellular CoA synthesis and/or degradation. As described in the introduction, the first three steps of the CoA synthesis are located in the cytoplasm, and the localization of the last two steps may be in the cytoplasm (on the outer mitochondrial membrane), in the mitochondrial matrix, or in both compartments [4]. Considering that the rate-limiting step in CoA biosynthesis is the phosphorylation of pantothenic acid localized in the cytoplasm [3,18] and that the hepatic mitochondrial CoA pool was maintained in BDL rats, hepatic CoA synthesis appears to function normally in BDL rats. The most likely explanation for the findings in the current study is, therefore, increased degradation of CoA and acyl-CoAs in the cytosol, e.g., by NUDT7 and NUDT19 in peroxisomes. Since peroxisomes can degrade CoASH and acyl-CoAs [4], the observation that there was no change in the distribution of the cytosolic CoA subfractions in BDL compared to control rats is compatible with this hypothesis. More detailed studies including the direct determination of CoA synthesis and degradation are necessary, however, to prove this assumption.

Recent studies have shown that CoASH can form a covalent, reversible binding with the mercapto group of cysteine residues in proteins called protein CoAthiolation [19,20,21]. Protein CoAthiolation protects proteins from oxidative damage but can potentially also decrease the cellular CoA pool, particularly in case of oxidative stress. Since previous studies have shown that bile duct ligation is associated with hepatocellular oxidative stress [22], protein CoAthiolation could be an additional mechanism that decreases the cytosolic CoA pool in the hepatocytes of BDL rats.

Surprisingly, we observed no significant decrease in the N-acetylation of sulfamethoxazole in BDL compared to control rats, despite the low hepatic cytosolic SCA-CoA content in BDL rats. The N-acetylation of arylamines such as sulfamethoxazole is a mainly hepatic cytosolic reaction in humans and rats [23,24,25]. In rats, the K_m_ for acetyl-CoA regarding this reaction has been reported to be 50 µmol/L [26], which is in the same range as the estimated acetyl-CoA concentration in BDL rats in the hepatocellular cytoplasm. Considering also that CoA biosynthesis can respond rapidly to drops in the cellular CoA concentration [4], the maintenance of the hepatic N-acetylation capacity in BDL rats is explainable. We also investigated the question of whether the reduced cytosolic CoASH concentration can become limiting for certain metabolic pathways. One of these pathways is long-chain fatty acid metabolism, of which the first step (fatty acid activation) occurs mainly on the mitochondrial outer membrane [27,28]. In rodents and humans, there are five long-chain acyl-CoA synthetase (ACSL) isoforms, of which ACSL1 and ACSL5 are important for the activation of long-chain fatty acids in hepatocytes. The results in Table 3 show that CoASH, at the concentration expected in the hepatocyte cytoplasm of BDL rats, does not limit palmitate activation, which is compatible with the K_m_ values for CoA reported for ACSL1 (6.4 µM) and ACSL5 (2.4 µM) [29]. The observation that palmitate activation by isolated mitochondria was not impaired in BDL compared to control rats suggests that the activity of ACSL1 and ACSL5 is maintained in BDL rats and that the reduction in palmitate activation in liver homogenate of BDL rats is explained by the known decrease in the hepatocyte volume fraction [12].

In contrast to the N-acetylation of sulfamethoxazole, hippurate formation from benzoic acid was decreased in BDL rats. A non-significant decrease in the same order of magnitude found in the current study has been reported by us in a previous study [30]. In the previous study, the difference in urinary hippurate excretion did not reach statistical significance due to high variability in BDL rats. Since the mitochondrial CoASH content (expressed per mg of mitochondrial protein and per total liver) was maintained in BDL rats, a decrease in mitochondrial CoASH is not an explanation for this finding. For the formation of hippurate, benzoic acid must be activated in a reaction needing ATP, which is a mitochondrial function. As shown in Table 4, and as reported already in previous studies [11,31,32], the oxidative metabolism of glutamate and succinate is decreased in mitochondria from BDL rats due to an impaired function of the mitochondrial electron transport chain. Reduced mitochondrial oxygen consumption and maintained ADP/O ratio in mitochondria from BDL rats suggest impaired mitochondrial ATP production at least under state 3 conditions. The impaired production of acid-soluble products from palmitate, despite the maintained palmitate activation by liver mitochondria in BDL rats, supports the notion that liver mitochondrial function is hampered in BDL rats. Impaired mitochondrial function is a possible explanation for the observed reduction in hippurate formation by BDL rats.

In conclusion, rats with long-term bile duct ligation have a reduced hepatocellular cytosolic CoA content, while the mitochondrial CoA content is maintained. The reduction in the cytosolic acetyl-CoA content is not large enough, however, to impair the N-acetylation of sulfamethoxazole and the activation of palmitate. A possible explanation for the reduction in the hepatocellular cytosolic CoA content in BDL rats is an increase in the peroxisomal degradation of CoA and acyl-CoAs.

## 4. Materials and Methods

### 4.1. Reagents

All reagents used were obtained at the highest purity available from the same suppliers as described in the original publications. 1-^14^C-palmitate was purchased from Parkin-Elmer (Schwerzenbach/Switzerland).

### 4.2. Animals

The animal experiments were reviewed and accepted by the Animal Ethics Committee of the state of Berne/Switzerland. Male Sprague-Dawley rats were obtained from the Süddeutsche Zuchttierfarm (Tuttlingen, Germany). Rats were kept on a 12 h dark/light cycle and had free access to food (Kliba Futter, Basel/Switzerland) and water. The ligation and transection of the common bile duct as well as the sham operation were performed as described previously [10]. The rats were characterized by the plasma concentration of bilirubin and the activities of AST and alkaline phosphatase (COBAS analyzer, Roche Diagnostics, Basel/Switzerland) and by the plasma concentration of bile acids (Becton Dickinson, Orangeburg, SC, USA). The experiments were performed with the rats in the fed state.

### 4.3. Fractionation of Liver Tissue

After bile duct ligation or sham operation for 28 days, rats were killed by stunning and decapitation. For the determination of metabolite concentrations and enzyme activities, mixed blood was obtained from the trunk into heparinized tubes. The fractionation of the livers was performed as described by Hoppel et al. [33]. In brief, livers were rapidly excised and washed with ice-cold MSM solution (220 mmol/L mannitol, 70 mmol/L sucrose, and 5 mmol/L 3-N-Morpholinolpropanesulfonic acid (MOPS), pH 7.4). All subsequent steps were performed at 5 °C. Approximately 6 g of liver tissue was minced, washed with MSM to remove blood clots, and homogenized in MSM containing 8 mmol/L EDTA (homogenate 1:10 wt:vol) using a Potter–Elvehjem Teflon/glass homogenizer. The resulting homogenate (liver homogenate) was centrifuged at 700× *g* for 10 min. The pellet (containing cell debris and nuclei) was discarded, and the supernatant (containing cytoplasm and cell organelles) was centrifuged at 7600× *g* for 10 min to isolate the mitochondria. The resulting pellet (containing a heavy and a light fraction) was shaken (to remove the light fraction which contains mainly lysosomes and peroxisomes) and the tube was wiped with gauze (to remove fat), resuspended in MSM, and washed twice using the same procedure. All 7600× *g* supernatants were pooled (cytosolic fraction) and the final pellet (approximately 150 mg mitochondrial protein) was collected (mitochondrial fraction). Liver homogenate, cytosolic, and mitochondrial fractions were stored at −80 °C until analysis. These fractions were analyzed for the activity of citrate synthase (mitochondrial marker) and lactate dehydrogenase (cytosolic marker). Citrate synthase activity was determined according to Srere [34] and lactate dehydrogenase activity according to Vassault [35]. Protein concentrations were determined according to Lowry [36] using bovine serum albumin as a standard. The activities of lactate dehydrogenase and citrate synthase were used to correct for losses concerning the cytosolic and mitochondrial fractions, respectively, during the isolation procedure.

### 4.4. Determination of the CoA Content in Liver Homogenate and Liver Subcellular Fractions

The work-up of the liver fractions was performed as described before [30]. Briefly, 100 µL of the liver homogenate or cytosolic fraction and 30 µL of mitochondrial fraction (approximately 5 mg of mitochondrial protein) were mixed with 20 µL of 200 mmol/L dithiothreitol and 1.88 mL (homogenate and cytosolic fraction) or 1.95 mL (mitochondrial fraction) of 3% perchloric acid (wt:vol). The suspensions were vortexed, kept on ice for 5 min, and then centrifuged at 10,000× *g* for 10 min. The supernatant was removed and adjusted to 2 mL with 3% perchloric acid and analyzed immediately for CoASH and total acid-soluble CoA (TAS-CoA). The pellet was washed once with 3% perchloric acid and stored at −80 °C until analysis. For the determination of CoASH, the supernatant was neutralized with 2 mol/L K_2_CO_3_ solution (0.9 mL of supernatant plus 0.3 mL of K_2_CO_3_ solution). After centrifugation at 10,000× *g* for 2 min, the CoA concentration was determined using the radioenzymatic method described by Cederblad et al. [37] with the modifications described previously [38]. For the determination of TAS-CoA, the supernatant was hydrolyzed at room temperature for 30 min (0.5 mL of supernatant plus 0.3 mL of 1 mol/L KOH) and then neutralized with 0.1 mL of 1 mol/L MOPS, pH 7. After centrifugation at 10,000× *g* for 2 min, the CoA concentration was determined as described above. For the determination of the long-chain acyl-CoA (LCA-CoA) concentration, the pellet was thawed, mixed with 0.8 mL of 0.5 mol/L KOH, and heated at 65 °C for 1 h. The suspension was neutralized with 0.3 mL of 1 mol/L MOPS in 0.4 mol/L HCl. After centrifugation at 10,000× *g* for 5 min, CoA was determined in the supernatant as described above. The short-chain acyl-CoA (SCA-CoA) concentration (containing mainly acetyl-CoA and other acyl-CoAs up to an acyl-chain length of approximately 8 [10]) was calculated as the difference of TAS-CoA minus CoASH and the total CoA was calculated as the sum of TAS-CoA and LCA-CoA.

### 4.5. Metabolism of Palmitate by Liver Homogenate and Liver Mitochondria

The oxidation of 1-^14^C-palmitate by liver homogenate or isolated mitochondria was investigated as reported previously [39]. Isolated rat liver mitochondria (250 µg mitochondrial protein) or 50 µL of rat liver homogenate (corresponding to 5 mg of liver tissue) were preincubated for 10 min at 37 °C in a final volume of 450 µL containing (final assay concentrations) 70 mM sucrose, 43 mM KCl, 3.6 mM MgCl_2_, 7.2 mM KH_2_PO_4_, 36 mM Tris, 2 mM ATP, 500 µM L-carnitine, 5 mM acetoacetate (pH 7.4), and CoA, as indicated in Section 2. The reaction was initiated by the addition of 50 µL of radioactive substrate (100 µM palmitate and 100 µM fatty-acid-free BSA, both final concentrations, and 5 nCi 1-^14^C-palmitate). The reactions were stopped after 5 min by adding 100 µL of 20% perchloric acid and radioactivity was determined in the supernatant by liquid scintillation counting after centrifugation (7000× *g* for 2 min). The results are reported as 1-^14^C-palmitate consumed.

The activity of palmitate activation was determined according to Reinartz et al. [40] with some modifications. In brief, isolated rat liver mitochondria (250 µg of mitochondrial protein) or 50 µL of rat liver homogenate (5 mg of liver tissue) was preincubated for 10 min at 37 °C in a final volume of 450 µL containing (final assay concentrations) 20 mM ATP, 200 µM CoA, 1 mM EDTA, 1 mM DTT, 5 mM MgCl_2_, 0.1% Triton X-100, and 150 µM Tris, pH 7.5. The assay was initiated by the addition of 50 µL radioactive substrate (100 µM palmitate and 100 µM fatty-acid-free BSA, both final concentrations, and 5 nCi 1-^14^C-palmitate). The reaction was stopped after 5 min by the addition of 1 mL of Dole’s medium followed by phase separation [41]. The water phase was counted for radioactivity by liquid scintillation.

### 4.6. In Vivo Metabolism of Benzoate

The metabolism of benzoate was determined according to Ahern and Mitchell [42] with some modifications, as described previously [30]. Briefly, a 200 mmol/L benzoic acid solution was prepared in distilled water and adjusted to pH 7 with NaOH. From this solution, 500 µL (100 µmoles) per 100 g body weight was injected intraperitoneally and the urine was collected in individual portions from 0 to 4 h, 4 to 8 h, 8 to 12 h, and 12 to 24 h. After the determination of the urine volumes, an aliquot was frozen and kept at −80 °C until analysis. The hippurate concentration was determined in the urine samples using the HPLC method of Arin et al. [43]. Precision (coefficient of variation of successive daily determinations) and accuracy (mean difference between actual and observed values) were both <10%.

### 4.7. In Vivo Metabolism of Sulfamethoxazole

Arylamines such as sulfamethoxazole are acetylated by N-acetyltransferases located mainly in the cytoplasm of hepatocytes [23]. The N-acetylation of sulfamethoxazole has therefore been used to investigate the hepatic cytosolic pool of acetyl-CoA [44]. To assess the metabolism of sulfamethoxazole, we prepared a solution of 100 mg/mL in NaCl 0.9%. From this solution, we injected 0.1 mL (10 mg) per 100 g body weight intraperitoneally into rats and collected the urine from 0 to 4 h, 4 to 8 h, 8 to 12 h, and 12 to 24 h. After having recorded the volume of the urine portions, we stored aliquots at −80 °C until analysis. The concentrations of sulfamethoxazole and N-acetyl-sulfamethoxazole were determined by HPLC using the method of Weber et al. [45]. The coefficient of variation of successive daily samples was <8% and the mean difference between actual and observed values was <10% for both analytes.

### 4.8. Statistical Analysis

Data are presented as the mean ± SEM. Statistical analyses were performed using GraphPad Prism 5 (GraphPad Software, La Jolla, CA, USA). Differences between the two groups were tested using Student’s *t*-test. The effect of bile duct ligation and CoA concentration on palmitate activation and acid-soluble product formation was analyzed by 2-way ANOVA followed by the Holm–Sidak procedure to localize differences. A *p*-value < 0.05 was considered statistically significant.

## Figures and Tables

**Figure 1 ijms-24-04365-f001:**
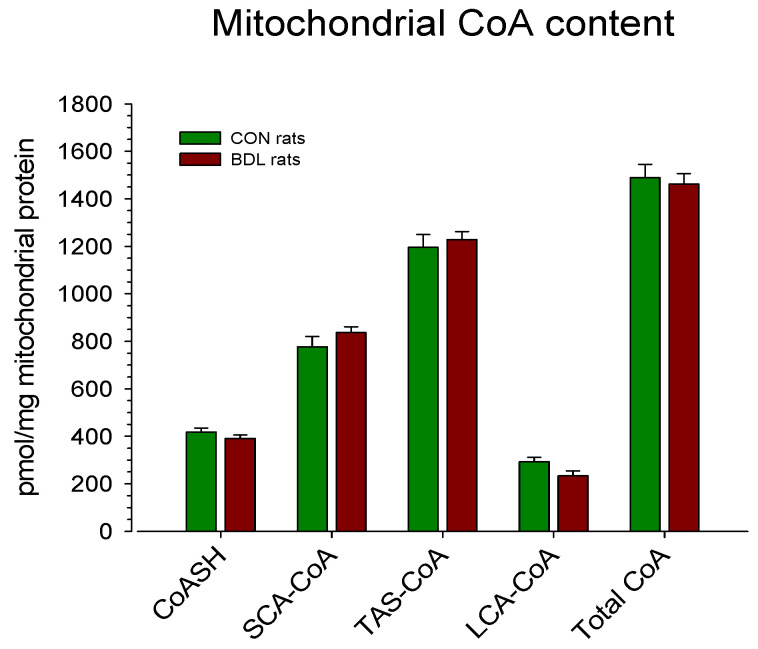
Mitochondrial CoA content. Rats were bile duct ligated for 4 weeks (BDL rats, *n* = 9) or sham-operated control rats (*n* = 5). After euthanasia by cervical dislocation, the liver was removed and homogenized. Mitochondria were isolated from the homogenate by serial centrifugation. The CoA content was determined in isolated mitochondria using a radioenzymatic method, as detailed in Methods. Data are presented as the mean ± SEM.

**Figure 2 ijms-24-04365-f002:**
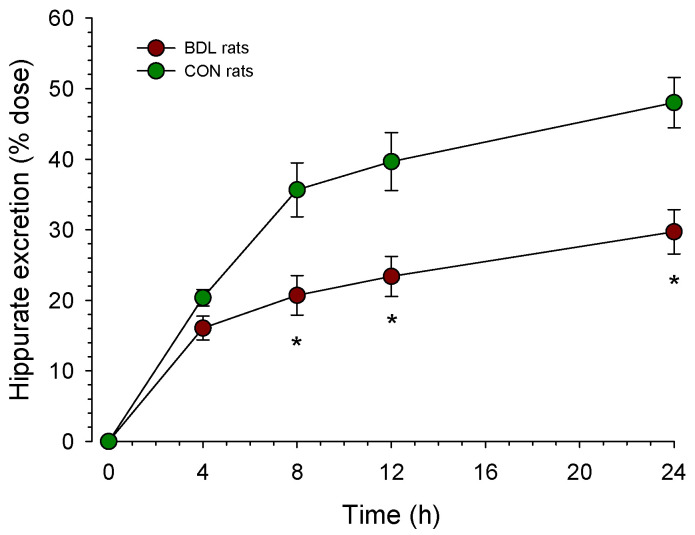
Urinary excretion of hippurate after the administration of benzoate. Rats were bile duct ligated for 4 weeks (BDL rats, *n* = 9) or sham-operated control rats (*n* = 5). After the i.p. administration of 100 µmoles of benzoate per 100 g of body weight, urine was collected in individual portions from 0 to 4 h, 4 to 8 h, 8 to 12 h, and 12 to 24 h. The hippurate concentration was determined by HPLC, as described in Methods. Data are presented as mean ± SEM. * *p* < 0.05 vs. CON rats.

**Figure 3 ijms-24-04365-f003:**
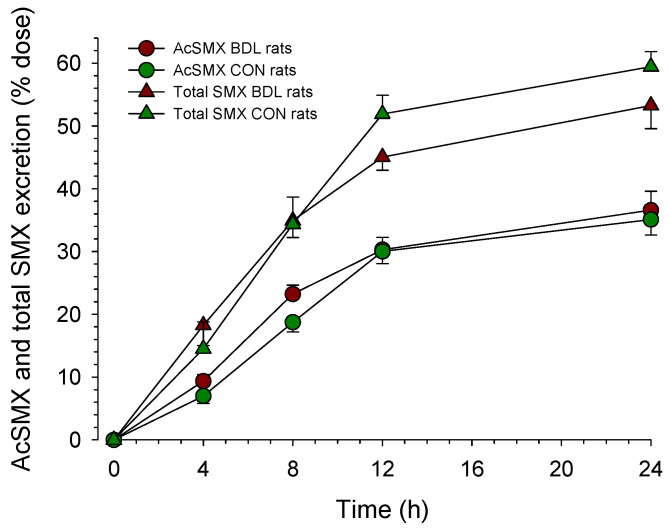
Urinary excretion of N-acetyl-sulfamethoxazole (AcSMX) after the administration of sulfamethoxazole (SMX). Rats were bile duct ligated for 4 weeks (BDL rats, *n* = 9) or sham-operated control rats (*n* = 5). After the i.p. administration of 10 mg of SMX per 100 g body weight, urine was collected in individual portions from 0 to 4 h, 4 to 8 h, 8 to 12 h, and 12 to 24 h. The SMX and AcSMX concentrations in urine were determined by HPLC, as described in Methods. Data are presented as mean ± SEM.

**Table 1 ijms-24-04365-t001:** Characterization of the animals. Rats were bile duct ligated for 4 weeks (BDL rats, *n* = 9) or sham-operated control rats (*n* = 5). They were studied in the fed state. Data are presented as the mean ± SEM. * *p* < 0.05 vs. control.

	Control (*n* = 5)	BDL Rats (*n* = 9)
Body weight (g)	339 ± 9	356 ± 8
Liver weight (g/100 g body weight)	3.69 ± 0.12	7.71 ± 0.15 *
Spleen weight (g/100 g body weight)	0.20 ± 0.01	0.68 ± 0.02 *
Serum bilirubin (µmol/L)	1 ± 1	114 ± 4 *
Serum bile acids (µmol/L)	1 ± 1	31 ± 3 *
AST (U/L)	72 ± 3	147 ± 6 *
Alkaline phosphatase (U/L)	203 ± 20	460 ± 10 *

**Table 2 ijms-24-04365-t002:** CoA content per g of liver wet weight in liver homogenate, cytoplasm and mitochondria.

	Control (*n* = 5)	BDL Rats (*n* = 9)
Liver homogenate		
CoASH	43.9 ± 1.8	30.3 ± 1.3 *
SCA-CoA (% total CoA)	143 ± 6 (68 ± 3)	81.4 ± 2.1 * (64 ± 3)
TAS-CoA	187 ± 8	112 ± 3 *
LCA-CoA	23.2 ± 0.4	16.0 ± 0.3 *
Total CoA	210 ± 9	128 ± 5 *
Liver mitochondria		
CoASH	31.4 ± 1.2	26.2 ± 1.0 *
SCA-CoA (% total CoA)	58.2 ± 3.2 (52 ± 4)	56.2 ± 1.6 (57 ± 3)
TAS-CoA	89.6 ± 4.0	82.4 ± 2.3
LCA-CoA	21.9 ± 1.4	15.7 ± 1.4 *
Total CoA (% total liver)	112 ± 5 (53 ± 4)	98.1 ± 3.0 * (77 ± 4 *)
Liver cytoplasm		
CoASH	20.8 ± 3.0	4.18 ± 0.67 *
SCA-CoA (% total CoA)	62.8 ± 3.2 (74 ± 4)	18.6 ± 0.6 * (81 ± 3)
TAS-CoA	83.6 ± 3.5	22.8 ± 0.8 *
LCA-CoA	1.04 ± 0.13	0.08 ± 0.03 *
Total CoA (% total liver)	84.6 ± 3.7 (41 ± 2)	23.0 ± 0.9 * (18 ± 1 *)

Abbreviations: CoASH: free CoASH; SCA-CoA: short-chain acyl-CoA; TAS-CoA: total acid-soluble CoA; LCA-CoA: long-chain acyl-CoA; total-CoA: TAS-CoA + LCA-CoA. Rats were bile duct ligated for 4 weeks (BDL rats, *n* = 9) or sham-operated control rats (*n* = 5). After euthanasia by cervical dislocation, the liver was removed and homogenized. After homogenization, the liver was fractionated into cytoplasm and mitochondria by serial centrifugation, as described in Methods. The CoA content was determined using a radioenzymatic method, as detailed in Methods. Units are nmol/g of wet weight. Data are presented as the mean ± SEM. * *p* < 0.05 vs. control.

**Table 3 ijms-24-04365-t003:** Palmitate metabolism by liver homogenate and isolated liver mitochondria. Rats were bile duct ligated for 4 weeks (BDL rats) or sham-operated control rats and were studied in the fed state. The formation of palmitoyl-CoA and or acid-soluble products from 1-^14^C-palmitate was determined as described in Methods. The estimated CoASH concentration in the assays with homogenate without the addition of CoASH was 0.2 µM for control and 0.04 µM for BDL rats. The final CoASH concentration in assays with the addition of exogenous CoASH is given in parentheses. Data are presented as the mean ± SEM. The % difference in the activity between BDL and control rats is given in parentheses. Units are nmol/min/g of liver wet weight for homogenate and nmol/min/mg mitochondrial protein for isolated mitochondria. * *p* < 0.05 vs. control rats and ^+^
*p* < 0.05 vs. respective homogenate incubations at 150 µM CoA.

	Control (*n* = 5)	BDL Rats (*n* = 9)
Formation of palmitoyl-CoA
Homogenate (no exogenous CoASH)	48.2 ± 3.1 ^+^	8.1 ± 1.2 *^+^ (83%)
Homogenate (5 µM CoASH)	152 ± 11	98.0 ± 5.1 * (36%)
Homogenate (150 µM CoASH)	161 ± 10	102 ± 6 * (37%)
Mitochondria (150 µM CoASH)	2.65 ± 0.15	2.58 ± 0.17
Formation of acid-soluble products
Homogenate (no exogenous CoASH)	25.4 ± 1.9 ^+^	4.7 ± 1.1 *^+^ (81%)
Homogenate (5 µM CoASH)	75.2 ± 5.2	25.6 ± 2.1 * (66%)
Homogenate (150 µM CoASH)	81.9 ± 4.8	30.1 ± 2.2 * (63%)
Mitochondria (150 µM CoASH)	1.72 ± 0.09	1.21 ± 0.7 * (30%)

**Table 4 ijms-24-04365-t004:** Mitochondrial metabolism of glutamate and succinate. Rats were bile duct ligated for 4 weeks (BDL rats) or sham-operated control rats and were studied in the fed state. Mitochondria were isolated by differential centrifugation, as described in Methods. State 3 oxidation rates are expressed as natoms/min/mg mitochondrial protein. RCRs (respiratory control ratios) and ADP/O ratios were calculated as described in Methods. The percentage difference in the activity between BDL and control rats is given in parentheses. Data are presented as the mean ± SEM. * *p* < 0.05 vs. control rats.

	Control (*n* = 5)	BDL Rats (*n* = 9)
L-glutamate 20 mM		
State 3	95.6 ± 4.8	59.0 ± 3.1 * (38%)
RCR	7.3 ± 0.4	6.7 ± 0.5
ADP/O	2.9 ± 0.2	2.6 ± 0.2
Succinate 20 mM		
State 3	199 ± 14	128 ± 9 * (36%)
RCR	5.8 ± 0.4	4.9 ± 0.5
ADP/O	2.0 ± 0.2	1.9 ± 0.2

## Data Availability

The data that support the findings of this study are available from the corresponding author upon reasonable request.

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
