# Peer review of "Rats with Long-Term Cholestasis Have a Decreased Cytosolic but Maintained Mitochondrial Hepatic CoA Pool"

_ijms, 2023, doi:10.3390/ijms24054365_

Round 1
Reviewer 1 Report (Previous Reviewer 1)
Authors corrected article according to suggestions. I recommend to accept artcile
Author Response
Thank you for your comment.
Reviewer 2 Report (Previous Reviewer 2)
Unfortunately, I strongly agree with the Reviewer #3 for the rejection of this paper because of its overall low-quality and no impacts on the current science.
Author Response
We gave our best to answer the questions raised by Reviewer 2 in our first version of the answers to the Reviewers. Obviously, the Reviewer kept his or her mind that the paper should not be published. Unfortunately, the Reviewer did not comment our answers and the new data we provided. This was done by Reviewer 3 (please, look at the comments of Reviewer 3 and our answers) who obviously changed his/her mind after having read our reply. In this situation, we have done what we could do and we must await the decision of the Editor.
Reviewer 3 Report (Previous Reviewer 3)
I appreciate the authors' acknowledgement of the limitations of their study in not investigating the reasons for the decrease in the hepatocellular cytosolic CoA pool in BDL rats. However, I would like to suggest that in future studies, the authors consider investigating the potential causes of this decrease in order to provide a more comprehensive understanding of the topic. Additionally, I appreciate the inclusion of data on palmitate metabolism in liver homogenate and isolated liver mitochondria to further assess the cytosolic CoA pool. Overall, I believe the authors have provided valuable insights into the functional consequences of the suspected decrease in the hepatocellular cytosolic CoA pool in BDL rats.
I understand the authors' disagreement with my previous comment. However, I would suggest that the authors consider providing more evidence to support their findings in the previous report, as well as discussing the potential reasons for the lack of statistical significance in the numerical decrease in hippurate formation. Additionally, I appreciate the authors' demonstration of impaired hepatic mitochondrial function in BDL rats through their assessment of palmitate beta-oxidation and oxidative metabolism of glutamate and succinate and would recommend that the authors continue to provide further explanation and evidence to support these findings in future.
Author Response
Thank you for your comments. We would like to answer as follows:
I appreciate the authors' acknowledgement of the limitations of their study in not investigating the reasons for the decrease in the hepatocellular cytosolic CoA pool in BDL rats. However, I would like to suggest that in future studies, the authors consider investigating the potential causes of this decrease in order to provide a more comprehensive understanding of the topic.
Answer: as written in the publication, the aims of the study were to confirm that the cytosolic CoA content is decreased in hepatocytes of BDL rats and to investigate the functional consequences of this suspected decrease. We of course fully agree that the mechanisms leading to the observed decrease in the cytosolic CoA pool should be known. This is the subject of a study that is currently in the planning phase.
Additionally, I appreciate the inclusion of data on palmitate metabolism in liver homogenate and isolated liver mitochondria to further assess the cytosolic CoA pool. Overall, I believe the authors have provided valuable insights into the functional consequences of the suspected decrease in the hepatocellular cytosolic CoA pool in BDL rats.
Answer: Thank you.
I understand the authors' disagreement with my previous comment. However, I would suggest that the authors consider providing more evidence to support their findings in the previous report, as well as discussing the potential reasons for the lack of statistical significance in the numerical decrease in hippurate formation.
Answer: In the current study, we found a hippurate excretion of 30±9% (mean±standard deviation) of the administered dose per 24 h in BDL rats (n=9) and of 48±9% in control rats (n=5). In the previous investigation (Hepatology 1997;25:278-283), the corresponding numbers were 68±18% in BDL rats (n=6) and 89±10% in control rats (n=5). In both studies, BDL rats excreted less hippurate per 24 h, the corresponding decreases were 38% in the current and 24% in the previous investigation. In the current investigation, the difference between CON and BDL rats was statistically significant, but not in the previous investigation. The reason why difference in the previous investigation did not reach statistical significance was the high standard deviation in the BDL rats. With a standard deviation of 9 as in the current study, the difference would have been significant. What is surprising when comparing the results of the two investigations, is the difference in the amount excreted between the two studies. We checked the protocols of the determinations and found no obvious error in the measurements and calculations. Rats were studied both in the fed state and had received the same food. A possibility is the urine collection, which may have not been complete in the current compared to the previous investigation.
We have added a sentence on page 8 of the revised manuscript stating that the difference in hippurate excretion in the previous investigation was not significant due to a high variability in the BDL rats.
Additionally, I appreciate the authors' demonstration of impaired hepatic mitochondrial function in BDL rats through their assessment of palmitate beta-oxidation and oxidative metabolism of glutamate and succinate and would recommend that the authors continue to provide further explanation and evidence to support these findings in future.
Answer: In our view, hepatic mitochondrial dysfunction is established in BDL rats. We have shown that in the current study and in other studies, e.g. J Hepatol 1998;28:1000-1007; Hepatology 1997;25:278-283; Hepatology 1994;19:1272-1281; Hepatology 1992;15:1167-1172. We had to show it in the current study since Reviewer 2 was not convinced regarding liver mitochondrial dysfunction in the BDL rats investigated in the current study. Since we had tested the isolated mitochondria for quality control, we could provide these data.
Reviewer 4 Report (New Reviewer)
The research presented in this manuscript is a logical extension of the work in the same laboratory on investigating the level of total CoA in the rat liver after long-term cholestasis caused by bile duct ligation (BDL). Here, the pool of CoA and CoA thioesters was measured in total liver homogenates, cytosolic and mitochondrial fractions under various experimental conditions. It was found that the total level of CoA was lower in BDL, when compared to control liver samples. Interestingly, the mitochondrial level of total CoA was maintained, but the reduction was observed in cytosolic CoA pool. Administration of benzoate, sulfamethyloxazole and palmitate into experimental rats was used to examine the involvement of CoA/CoA derivatives in their metabolism and urinary excretion. Based on these findings, the authors speculated that a) the reduction of cytosolic pool of CoA in BDL rats is most likely associated with its degradation by Nudix enzymes; b) mitochondrial dysregulation leads to the impaired hippurate excretion after benzoate administration into BDL rats.
The research on CoA biology has had a reviving trend in the last two decades, promoted by molecular cloning of genes involved of the CoA biosynthetic pathway, appreciation of the antioxidant function of CoA and better understanding of dysregulated CoA homeostasis in human pathologies. CoA is expressed at high level in the liver, where it is involved in diverse pathways of cellular metabolism, as well as detoxification processes. Moreover, the level of CoA/CoA thioesters differs significantly in different subcellular compartments. Therefore, measuring how the level of CoA changes in the liver in pathologies, including bile duct obstruction, may provide insights for better understanding of liver degeneration, development of novel drug targets and therapies. This manuscript reveals how the tevel of total CoA changes in the cytosol and mitochondria in the rat liver after BDL, and therefore, it is timely and deserves to be published in IJMS, after the authors have responded to the comment below and made changes where necessary.
Bile duct obstruction is known to be associated with oxidative stress and mitochondrial dysfunction. CoA has been recently shown to function as a major cellular antioxidant in cellular response to oxidative stress. This function is known to be mediated by covalent protein modification of oxidised cysteine residues. Therefore, covalent protein modification by CoA may also contribute to the observed reduction in the level of total CoA in the liver after BDL. The authors should also consider this mechanism in the Discussion, alongside with other proposed explanations. Measuring the level of total CoA under reducing conditions (in the presence of DTT) might be the subject of further investigations in this exp model.
Author Response
Bile duct obstruction is known to be associated with oxidative stress and mitochondrial dysfunction. CoA has been recently shown to function as a major cellular antioxidant in cellular response to oxidative stress. This function is known to be mediated by covalent protein modification of oxidized cysteine residues. Therefore, covalent protein modification by CoA may also contribute to the observed reduction in the level of total CoA in the liver after BDL. The authors should also consider this mechanism in the Discussion, alongside with other proposed explanations. Measuring the level of total CoA under reducing conditions (in the presence of DTT) might be the subject of further investigations in this exp model.
Answer: Thank you for your comments and for this important remark. Indeed, we have shown previously that the hepatocellular content of antioxidants such as glutathione and ubiquinone is decreased in BDL rats and that thiobarbituric acid reacting substances are increased, suggesting oxidative stress (Hepatology 1995;22:607-612). Recent studies have shown that coenzyme A can bind to cysteine thiols and protect cysteine containing proteins from oxidative damage (reviewed in Biochem Soc Trans 2018;46:721-728). On the one hand, this is protective against oxidative stress, on the other hand, this mechanism could reduce the cellular CoA content. As suggested by the Reviewer, we comment these findings in the Discussion on page 8 of the revised manuscript.
This manuscript is a resubmission of an earlier submission. The following is a list of the peer review reports and author responses from that submission.
Round 1
Reviewer 1 Report
It is interesting article about CoA analysis in rats with cholestasis , I have just few comments:
1) Tha Tables in the text should have numbers and description as well as abbreviation under th table to easiest follow the results;
2) in discussion the practical aspects of this study should be added and discussed
Author Response
1) The Tables in the text should have numbers and description as well as abbreviation under the table to easiest follow the results
Answer: We have numbered the Tables and present them as suggested by the Reviewer.
2) in discussion the practical aspects of this study should be added and discussed
Answer: There are at least two practical consequences.
One of them is related to the finding that the contents of the cytosolic and the mitochondrial CoA pools do not change mutually. This underscores the tightness of the inner mitochondrial membrane towards CoASH and acyl-CoAs and suggests a very limited transport of CoASH/acyl-CoAs from the mitochondrial matrix into the cytoplasm. We have enlarged the discussion on page 9 on this point.
The second point relates to the question whether changes in the CoA pool affect enzymatic reactions depending on free CoA or acetyl-CoA. This is why we looked at the metabolism of benzoate, sulfamethoxazole and palmitate. Despite the drop in the cytosolic acetyl-CoA concentration, the acetyl-CoA concentration is still high enough to support N-acetylation of sulfamethoxazole. We also determined palmitate metabolism in vitro to test the question whether the CoA concentration in the cytosol is limiting for fatty acid activation. We found that this is not the case in CON and in BDL rats. Please consider our answer to Reviewer 2 regarding this point. We have enlarged the discussion in the manuscript regarding the cytosolic and mitochondrial CoA/acetyl-CoA pools (page 9).
Reviewer 2 Report
The authors had reported long ago (in 1991 [Ref.10] and 1994 [Ref.11]) that the hepatic CoA content was reduced while its “mitochondrial” content was maintained in BDL rats (lines 157–159), and this study (after 30 years!!) revealed that liver “cytosolic” CoA was markedly reduced. However, unexpectedly, they found that benzoate (in vivo) metabolism to hippurate (that uses Mito CoA) was 37% reduced whereas N-acetylation of sulfamethoxazole (that requires cytosolic CoA) was maintained without any mechanistic/molecular insights supported by experimental evidences (e.g.measurement of cytosolic CoA degradation (line 202), impaired ATP production (line 220)). Overall, this manuscript is quite premature in both scientific quality and English grammar, and does not provide enough information/novel findings that are needed for regular full-paper journals.
Some concerns:
1. In abstract
A) Line 10: “liver cytosolic” rather than “hepatocellular”
B) Line 11–12“determined the CoA pool” and “tested the …CoA pools” are redundant.
C) Line 13 (and line 64) : What does “functionally” mean?
2. Table 1 and 2 should be labelled.
3. Table 1 and Figs.1/2 are just redundant. Remove Figs.1/2.
4. There are many redundant phrases in the results and discussion.
Author Response
The authors had reported long ago (in 1991 [Ref.10] and 1994 [Ref.11]) that the hepatic CoA content was reduced while its “mitochondrial” content was maintained in BDL rats (lines 157–159), and this study (after 30 years!!) revealed that liver “cytosolic” CoA was markedly reduced. However, unexpectedly, they found that benzoate (in vivo) metabolism to hippurate (that uses Mito CoA) was 37% reduced whereas N-acetylation of sulfamethoxazole (that requires cytosolic CoA) was maintained without any mechanistic/molecular insights supported by experimental evidences (e.g.measurement of cytosolic CoA degradation (line 202), impaired ATP production (line 220)). Overall, this manuscript is quite premature in both scientific quality and English grammar and does not provide enough information/novel findings that are needed for regular full-paper journals.
Answer: we of course agree that the study is based on investigations performed 20 to 30 years ago. We clearly state this in the introduction and provide the corresponding references. The current study confirms these results and shows now directly that the cytosolic CoA pool in hepatocytes of BDL rats is reduced compared to control rats. This could be suspected from the available data but had, at least in our view, to be proven. The study focused on the consequences (not on the reasons) of this finding. That’s why we determined the N-acetylation of sulfamethoxazole and the formation of hippurate from benzoate. Since we originally tested only the acetyl-CoA pool in the cytoplasm, we added an in vitro investigation regarding the cytosolic CoA pool. Palmitate is activated (formation of palmitoyl-CoA) mainly on the outer mitochondrial membrane and in the ER; activation is needed that palmitate can undergo β-oxidation or can be used for the formation of triglycerides. For that, we investigated activation of palmitate and the formation of acid soluble products from palmitate (β-oxidation) using liver homogenate without added CoASH, in the presence of the estimated CoASH concentration in the hepatocyte cytoplasm (5 µM) and at a very high CoASH concentration (150 µM). The results showed that without addition of exogenous CoASH, palmitate activation and acid soluble product formation is lower than after CoASH addition. However, there was no significant difference in palmitate activation or acid soluble product formation between 5 and 150 µM palmitate. Palmitate activation and acid soluble product formation were lower in homogenate from BDL than CON rats, explained by a loss of hepatocytes per g liver in BDL rats and, only for acid soluble product formation, mitochondrial dysfunction. These results suggest that the CoASH content in the hepatocellular cytoplasm of BDL rats (and control rats) does not limit palmitate activation and that liver mitochondrial function is impaired in BDL rats. We showed the impairment of liver mitochondrial in BDL rats also directly by assessing the oxidative metabolism of glutamate and succinate. We show the results of these studies in new Table 3 and 4 and have added the appropriate discussion of the new results.
We agree with the Reviewer that we speculate about the mechanism of the decrease in the cytosolic CoA/acyl-CoA content. We admit that further studies are needed to directly proof that CoA degradation is increased in hepatocytes of BDL rats. However, the focus of the study was different; as stated above, we were interested in the functional consequences of the decrease in hepatic CoA in BDL rats.
Some concerns:
- In abstract
- A) Line 10: “liver cytosolic” rather than “hepatocellular”
- B) Line 11–12“determined the CoA pool” and “tested the …CoA pools” are redundant.
- C) Line 13 (and line 64): What does “functionally” mean?
Answer: We have performed the requested changes. The demonstration that the cytosolic CoA content is reduced was one of the aims of the current project, this could be suspected but was not known before. The Reviewer may have misunderstood that. We have added a sentence to explain “functionally”.
- Table 1 and 2 should be labelled.
Answer: we have done that.
- Table 1 and Figs.1/2 are just redundant. Remove Figs.1/2.
Answer: we agree regarding Fig. 1 but not regarding Fig. 2. Figure 2 shows the CoA content per mg mitochondrial protein, which is not displayed in the Tables.
- There are many redundant phrases in the results and discussion.
Answer: we have checked the Discussion and removed redundant phrases as suggested.
Reviewer 3 Report
This manuscript “Rats with long-term cholestasis have a decreased cytosolic but maintained mitochondrial hepatic CoA pool” not fits well with the Aims and Scope of the International Journal of Molecular Science. In this manuscript the authors determined the CoA pool in liver homogenate, liver mitochondria and liver cytosol of rats with bile duct ligation and sham-operated control rats and tested the cytosolic and mitochondrial CoA pools functionally. They concluded that BDL rats have reduced hepatocellular cytosolic CoA stores but no impairment of sulfamethoxazole N- 23 acetylation. In comparison, the hepatocellular mitochondrial CoA pool is maintained in BDL rats. Thus, impaired hippurate formation in BDL rats is not due to lacking free CoA but most likely due to mitochondrial dysfunction. This manuscript has been written well, however there are some major problems in this manuscript.
1. The manuscript well described the phenomenon that BDL rats have reduced cytosolic CoA stores but maintained the mitochondrial CoA pool but lack the scientific aspect. The authors should explore the mechanism of this phenomenon.
2. The results in this manuscript are the opposite of those in another paper “Benzoic Acid Metabolism Reflects Hepatic Mitochondrial
Function in Rats with Long-Term Extrahepatic Cholestasis” by the authors. The authors should discuss this in the discussion.
Author Response
This manuscript “Rats with long-term cholestasis have a decreased cytosolic but maintained mitochondrial hepatic CoA pool” not fits well with the Aims and Scope of the International Journal of Molecular Science. In this manuscript the authors determined the CoA pool in liver homogenate, liver mitochondria and liver cytosol of rats with bile duct ligation and sham-operated control rats and tested the cytosolic and mitochondrial CoA pools functionally. They concluded that BDL rats have reduced hepatocellular cytosolic CoA stores but no impairment of sulfamethoxazole N- 23 acetylation. In comparison, the hepatocellular mitochondrial CoA pool is maintained in BDL rats. Thus, impaired hippurate formation in BDL rats is not due to lacking free CoA but most likely due to mitochondrial dysfunction. This manuscript has been written well, however there are some major problems in this manuscript.
- The manuscript well described the phenomenon that BDL rats have reduced cytosolic CoA stores but maintained the mitochondrial CoA pool but lack the scientific aspect. The authors should explore the mechanism of this phenomenon.
Answer: as written in the introduction, the aims of the study were 1. To prove the suspected decrease in the cytosolic CoA pool of hepatocytes in BDL rats and 2. To assess the function of the cytosolic and the mitochondrial CoA pools. So, we focused on the functional consequences of the suspected decrease in the hepatocellular cytosolic CoA pool in BDL rats and not on the reason for this decrease. We state the aims of the study clearly in the Introduction. We of course admit that lack of investigation of the reasons for the drop in the hepatocellular cytosolic CoA content in BDL rats is a weakness of the study which should be investigated in further studies.
To complete the assessment of the cytosolic CoA pool, we added data regarding palmitate metabolism in liver homogenate and isolated liver mitochondria. The results suggest that the cytosolic CoA pool is not limiting palmitate activation in BDL rats (please, consider our answer to Reviewer 2).
- The results in this manuscript are the opposite of those in another paper “Benzoic Acid Metabolism Reflects Hepatic Mitochondrial Function in Rats with Long-Term Extrahepatic Cholestasis” by the authors. The authors should discuss this in the discussion.
Answer: We disagree with the Reviewer in this point. Also in our previous report (Hepatology 1997;25:278-283) we found a numerical decrease in hippurate formation from i.p. administered benzoate in BDL compared to control rats, which, however, did not reach statistical significance. The numerical decrease was comparable to the current study, but the variability of the measurements was larger. We discuss that on page 9 of the manuscript. We also show directly that hepatic mitochondrial function is impaired in BBDL compared to control rats by assessing palmitate β-oxidation and oxidative metabolism of glutamate and succinate.